# 4D Spatio-Temporal Deep Learning with 4D fMRI Data for Autism Spectrum Disorder Classification

**Marcel Bengs**[*1]                                        MARCEL.BENGS@TUHH.DE

**Nils Gessert**[*1]                                         NILS.GESSERT@TUHH.DE

**Alexander Schlaefer**[1]                                   SCHLAEFER@TUHH.DE

[1]*Institute of Medical Technology, Hamburg University of Technology, Germany*

## Abstract

Autism spectrum disorder (ASD) is associated with behavioral and communication problems. Often, functional magnetic resonance imaging (fMRI) is used to detect and characterize brain changes related to the disorder. Recently, machine learning methods have been employed to reveal new patterns by trying to classify ASD from spatio-temporal fMRI images. Typically, these methods have either focused on temporal or spatial information processing. Instead, we propose a 4D spatio-temporal deep learning approach for ASD classification where we jointly learn from spatial and temporal data. We employ 4D convolutional neural networks and convolutional-recurrent models which outperform a previous approach with an F1-score of 0.71 compared to an F1-score of 0.65.

**Keywords:** 4D Deep Learning, 4D CNN, fMRI, ASD

## 1. Introduction

Autism spectrum disorder (ASD) is typically associated with difficulties in communication, repetitive behavior and restricted interests. The developmental disorder leads to life-long disability (Howlin et al., 2004). To understand the underlying causes and find treatments, functional magnetic resonance imaging (fMRI) has been employed which has lead to potential biomarkers of the disease (Loth et al., 2016). For example, functional connectivity computed from resting-state fMRI has been used to derived features for ASD classification with classical machine learning approaches such as support vector machines (Plitt et al., 2015). Recently, deep learning approaches have been proposed for ASD classification (Wen et al., 2018). For example, Dvornek et al. have used long short-term memory (LSTM) cells with sequences extracted from fMRI data (Dvornek et al., 2017). The input sequences were computed as the average signal over 200 manually selected regions of interest for each time point in the 4D fMRI image. Instead of using this handcrafted feature approach, Li et al. (Li et al., 2018) directly learned spatial features from the fMRI data using 3D convolutional neural networks (CNNs). However, this approach only considered temporal information implicitly by taking the mean and standard deviation volume over a fixed time window. Instead, we propose to directly learn from the full 4D fMRI sequences. This requires 4D spatio-temporal deep learning techniques which are largely unexplored so far. We employ

---

[*] Contributed equally

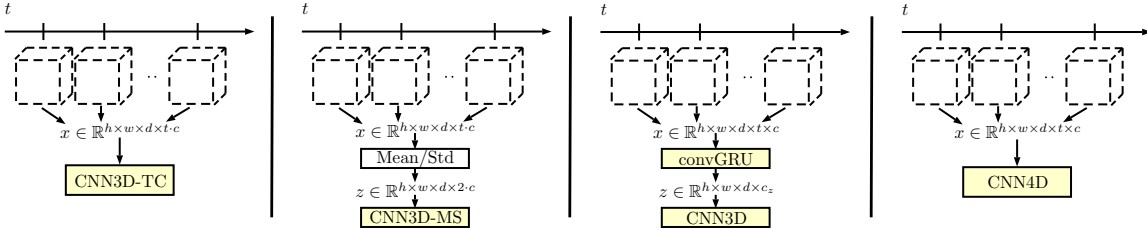

Figure 1: The four architectures we employ. Note that the images' color dimension is $c = 1$.

4D CNNs which process both spatial and temporal dimensions with convolutions. Moreover, we adopt a convolutional-gated recurrent unit 3D CNN (convGRU-CNN3D), which performs temporal processing first with subsequent spatial processing (Gessert et al., 2018). We compare our methods to the previous approach of using 3D CNNs with mean and standard deviation volumes derived from the 4D image (Li et al., 2018). Also, we consider a typical approach from the natural image domain where time steps are stacked in the channel dimension (Pfister et al., 2014). Summarized, we propose 4D spatio-temporal deep learning for ASD classification from 4D fMRI images. We compare to previous, mostly spatial approaches by employing four models with different ways of processing the temporal information.

## 2. Methods

**Dataset.** We use the publicly available ABIDE dataset (Craddock et al., 2013) which contains preprocessed resting-state 4D fMRI sequences of patients with ASD and healthy controls. We use a subset of 184 patients from the NYU collection site. The sequences are preprocessed with the Connectome Computation System pipeline which includes slice timing correction, motion realignment, intensity normalization and rigid registration to a template. Following (Dvornek et al., 2017), we employ bandpass filtering (0.01 - 0.1 Hz) and no global signal regression. Each 4D fMRI image is spatially downsampled to reduce computationally effort such that each fMRI image has a size of $32 \times 32 \times 32 \times 176$ where the first three dimensions are spatial and the last dimension is temporal. We split the dataset into a training, validation and test set with 134, 30 and 30 subjects, respectively. Validation and test set contain an equal number of ASD and control subjects.

**Models.** Our deep learning models receive a 4D image of size $32 \times 32 \times 32 \times 15$ as the input, cropped from the full 4D fMRI image. We employ models with four different types of temporal processing, see Figure 1. In general, all models follow the idea of densely connected convolutional networks (DenseNet) (Huang et al., 2017). The core architecture is a 3D CNN with one initial convolutional layer, followed by three DenseNet blocks with 5 layers each. After the final module we use a global average pooling, which is directly connected to the fully connected layer with two outputs for the two classes.

For the first type of temporal processing, we treat the CNN's input channel as the temporal dimension (Pfister et al., 2014). Thus, we stack the 15 volumes into the first layer's input

Table 1: Results for all experiments.

|  | CNN3D-TC | CNN3D-MS | convGRU-CNN3D | CNN4D |
|---|---|---|---|---|
| Accuracy | 0.57 | 0.60 | **0.67** | 0.60 |
| F1-score | 0.61 | 0.65 | **0.71** | 0.68 |

channel (CNN3D-TC).

Second, we compute the voxel-wise mean and standard deviation over the 15 volumes in a sequence which follows a previous approach for ASD classification (Li et al., 2018). Thus, we obtain two 3D images which we stack into the first layer's input channel (CNN3D-MS). Third, we learn a volume representation from the 15 time steps which is fed into the 3D CNN model. For this purpose, we use a gated recurrent neural network with convolutional gating operations (convGRU) (Xingjian et al., 2015) in front of our core DenseNet architecture (Gessert et al., 2018). The convGRU first performs temporal processing while keeping the data structure intact for subsequent spatial processing by the 3D CNN. The model is trained end-to-end (convGRU-CNN3D).

Fourth, we employ 4D CNNs by isotropically extending the network's convolutional kernels by a fourth temporal dimension. In contrast to the two-stage setup in convGRU-CNN, the 4D CNN performs spatio-temporal processing throughout the entire network (CNN4D).

**Training and Evaluation.** During training, we randomly crop sequences of length 15 from the fMRI image of each subject. For validation and testing, we crop and evaluate sequences from fMRI images in a sliding window fashion and average the predictions over all crops of each subject. We train our models for 500 epochs with a batch size of 10. For optimization of the cross-entropy loss function, we employ the Adam optimizer. During training, we evaluate the performance on the validation set every 5 epochs and use the best model for the final evaluation on the test set.

## 3. Results and Discussion

We report the F1-score and accuracy for our experiments in Table 1. In general, the classification accuracy is not very high and in a similar range as previous results on ASD classification with different datasets (Dvornek et al., 2017). This underlines that ASD classification from fMRI is a challenging problem. In detail, convGRU-CNN3D shows the highest performance across both metrics, followed by CNN4D. This indicates that explicit learning on the temporal dimension is beneficial for ASD classification from fMRI images. Previously, an approach similar to CNN3D-MS had been used for ASD classification from fMRI images (Li et al., 2018). This spatial approach and also CNN3D-TC only implicitly process temporal information. Our more powerful 4D deep learning models with explicit temporal processing appear to be able to learn richer features from the fMRI data. Overall, we propose 4D deep learning models for ASD classification from 4D fMRI sequences. We demonstrate that a 4D spatio-temporal convGRU-CNN3D and CNN4D outperform a previous spatial approach that only implicitly considered temporal information. Our results indicate that 4D deep learning models could be beneficial for other learning problems with 4D fMRI data.

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
