# OpenReview forum: "4D Spatio-Temporal Deep Learning with 4D fMRI Data for Autism Spectrum Disorder Classification"
_MIDL.io/2019/Conference/Abstract — MIDL Abstract 2019_

### Official Review · AnonReviewer1 · 2019-05-01
**Needs stronger evaluation**

**Rating:** 2
**Confidence:** 3

**Review:**

The paper presents a new method which claims to outperform a previous approach. However, the evaluation is not strong enough to conclude that the method is better. The study uses a fixed data split for train, valid, and test set. This does not allow for the variance of the model to be accounted for in the evaluation. It is possible that if the model was trained on ~10-20 data splits the mean performance would not show a change between the different models.

Also, why is the hand crafted feature approach by Li et al., 2018 compared to as a baseline?

---

### Official Review · AnonReviewer2 · 2019-05-06

**Rating:** 4
**Confidence:** 3

**Review:**

ASD classification from fMRI. Compared 4 different architectures, including 3D, ConvGRU-Cnn3d, and 4D cnns. The finding is that an explicit learning over the temporal dimension helps, as convgru model performed the best. Could be a good discussion paper

---

### Decision · Program_Chairs · 2019-05-06
**Acceptance Decision**

Accept